# Global LiDAR land elevation data reveal greatest sea-level rise vulnerability in the tropics

A. Hooijer [1,2✉] & R. Vernimmen [3]

Coastal flood risk assessments require accurate land elevation data. Those to date existed only for limited parts of the world, which has resulted in high uncertainty in projections of land area at risk of sea-level rise (SLR). Here we have applied the first global elevation model derived from satellite LiDAR data. We find that of the worldwide land area less than 2 m above mean sea level, that is most vulnerable to SLR, 649,000 km² or 62% is in the tropics. Even assuming a low-end relative SLR of 1 m by 2100 and a stable lowland population number and distribution, the 2020 population of 267 million on such land would increase to at least 410 million of which 72% in the tropics and 59% in tropical Asia alone. We conclude that the burden of current coastal flood risk and future SLR falls disproportionally on tropical regions, especially in Asia.

[1] Deltares, Inland Water Systems Unit, P.O. Box 177 Delft, The Netherlands. [2] NUS Environmental Research Institute (NERI), National University of Singapore, 1 Engineering Drive, Singapore, Singapore. [3] Data for Sustainability, Axel, The Netherlands. ✉email: aljosja.hooijer@deltares.nl

According to the latest International Panel for Climate Change (IPCC) report[1] the question is no longer whether sea-level rise (SLR) will exceed 0.8 m, but rather whether this will happen by 2100 or beyond. Importantly for flood risk projections, IPCC[1] also states with a high level of confidence that extreme sea level events have increased substantially in recent decades, especially in tropical regions, and predicts that events that historically occurred once per century are highly likely to occur annually by 2100. At the same time, land surface subsidence (LSS) exceeding 2.5 mm yr$^{-1}$ is reported for most inhabited coastal lowlands globally, with rates in the tropics being over 0.5 mm yr$^{-1}$ in rural areas and well over 20 mm yr$^{-1}$ in urban areas (Supplementary Table 1), exacerbating or even exceeding the impacts of SLR. In most populated coastal regions, the combined effects of SLR and LSS will therefore likely exceed a relative SLR (RSLR) of 1 m by 2100. In response, policy makers and scientists are looking at adaptation options to reduce the impact of these changes. This requires accurate quantification of current and expected flood levels relative to the coastal land surface.

The confidence in coastal flood risk assessments and projections to date has been reduced by the low accuracy of the Global Digital Elevation Models (GDEMs) available[2–6]. These are created from satellite radar data that often measure land surface levels that are several meters above the actual ground surface[7,8] resulting in an underestimation of current and future flood risk[9–12]. For three validation areas presented by Vernimmen et al.[8], SRTM[13] data and the derived MERIT[14] and CoastalDEM[15] GDEMs, which are most commonly used in flood risk assessments, have at 0.05-degree resolution a best accuracy within 0.5 m for only 19.9% (SRTM) of land with a best mean absolute error of 1.26 m and Root Mean Square Error (RMSE) of 1.44 m (CoastalDEM)[8]. For this assessment we applied a new global digital terrain model (DTM) for coastal lowlands, GLL_DTM_v1, created at 0.05-degree resolution from ICESat-2 satellite LiDAR data available since 2018, that has much improved accuracy compared to existing GDEMs[8]. The GLL_DTM_v1 is accurate within 0.5 m for 85.2% of land below 2 m above mean sea level (+MSL) with a mean absolute error of 0.29 m and RMSE of 0.50 m. This allows an analysis of the flooding effect of a RSLR of 1 m at 68% confidence level[16], which is not possible with existing GDEMs that all have RMSEs over 1 m, as the rate of RSLR must be at least twice the vertical accuracy[16]. We focus on coastal land below MSL that is at immediate potential risk of flooding in the absence of coastal protection, and land below 2 m +MSL that was found by Syvitski et al.[17] to be most susceptible to major river floods and storm surges and in much of the world is already below high tide sea levels.

## Results

**Global distribution of coastal lowlands**. The global extent of land below 2 m +MSL as determined from GLL_DTM_v1 is 1.05 million km$^2$, with a range of 0.78–1.27 million km$^2$ at the 68% confidence level (Table 1), of which 0.65 million km$^2$ (62%) is in tropical regions (23.5 N – 23.5 S). At 0.32 million km$^2$, almost a third (31%) is in the tropical regions of Asia alone (Table 1, Fig. 1, Fig. 2), reflecting the long coastlines, large river deltas and numerous islands in that region. Of the 20 countries with over 12,000 km$^2$ of land below 2 m +MSL, nine are fully or partly (China) in tropical Asia, and only four countries are outside the tropics (Table 1, Fig. 1).

Compared to existing GDEMs, GLL_DTM_v1 indicates substantially greater land areas below 2 m +MSL globally and in most individual countries and at higher confidence given that area ranges at 68 and 95% confidence levels are much smaller (Fig. 3, Supplementary Data). The comparison is more varied for

land below MSL, with GLL_DTM_v1 indicating larger areas globally than SRTM and MERIT but sometimes smaller than CoastalDEM and TanDEM-X, with major differences observed between countries. For Vietnam, for example, GLL_DTM_v1 yields 6200 km$^2$ on land below MSL whereas this is 10,300/900/35,200/4,400 for SRTM/MERIT/CoastalDEM and TanDEM-X, respectively (Fig. 3, Supplementary Data).

**Patterns and trends in lowland population**. The 2020 global population estimate for coastal land below 2 m +MSL is 267 million (range 197–347 at 68% confidence level) of which the majority of 191 million (72%) is in the tropics and as much as 157 million (59%) in tropical Asia alone. Of the eight countries with over 10 million people living below 2 m +MSL, six are fully or partly in tropical Asia and the other two in tropical Africa. The 2020 global population number below 0 m +MSL is 35 million, with half (18 million) being in the tropics of which most (15 million) are in tropical Asia (Table 1).

Global average population increase between 2000 and 2020 on land situated below 2 m +MSL in 2020 is 1.3% yr$^{-1}$. In tropical coastal lowlands this is 1.4% yr$^{-1}$, due in large part to higher population growth rates above 3% yr$^{-1}$ in coastal zones of much of Africa, compared to 1.1% in temperate and boreal regions in general (Fig. 1) and 0.5% in the United States and the Netherlands (Table 1).

If RSLR (SLR + LSS) by 2100 reaches 1 m above the 2020 level and we conservatively assume there is no increase in coastal population compared to 2020, the global extent of coastal lands below 0 and 2 m +MSL is projected to increase to 0.52/1.46 million km$^2$ (up 295%/40%) affecting a population of 129/410 million (up 273%/54%) (Table 1). Substantial densely populated parts of many major delta areas will be below MSL (Fig. 4).

## Discussion

The tropics stand out as having the largest numbers globally, both in terms of coastal land area below 2 m +MSL and the number of people living on it. The greatest such areas and populations are in tropical Asia at 31 and 59% of the 2020 global total, respectively. We find that tropical America follows in terms of land area but not population, at 20%/3%, while the relatively limited coastal lowland areas and populations in tropical Africa (9%/10%) have the highest population growth rates (Table 1, Fig. 1). Consequently, whether measured by area, population size or population growth the burdens of RSLR are likely to fall disproportionally upon developing countries in the tropics that often have limited capacity to adapt. The population numbers affected would be higher if coastal population density is expected to increase, however future coastal population developments in tropical regions are uncertain, amongst others due to migration[18], and have therefore not been considered in this assessment.

While the tropical land area below 2 m +MSL is projected to decrease slightly as a proportion of the global total, from 62 to 61% after 1 m RSLR, the relative area of land below MSL is projected to increase substantially from 47 to 58% of the global total, by 2100 or before. The associated rate of increase in people living below MSL in the tropics is 404% (from 18 to 92 million) of which 70 million would be in tropical Asia alone, more than triple that in temperate regions (127%; 16 to 37 million).

The high vulnerability to RSLR of land and populations in coastal regions and deltas in especially tropical Asia has been noted before[17,19,20]. However, this analysis, applying a substantially more accurate global elevation model than was available to date, is the first to relate precise numbers for tropical area and population below 0 and 2 m +MSL, i.e. at most immediate risk, to global numbers.

**Table 1 Coastal lowland areas and populations below 2 and 0 m above mean sea level (+MSL), by 2020 and after 1 m relative sea-level rise (RSLR). Applying the GLL_DTM_v1 at 0.05-degree resolution, for countries ranked by land area below 2 m +MSL greater than 12,000 km². Population data from ref. 22.**

Column groups: columns 2–13 = **Land below 2 m +MSL** (Area [10³ km²]: 2020, % country area, Range 68%, +1 m RSLR, Range 68%, Increase 2020-2100 [%]; Population [million]: 2020, Range 68%, +1 m RSLR, Range 68%, Increase 2020-2100 [%], 2000-20 pop. growth [$yr^{-1}$]). Columns 14–24 = **Land below 0 m +MSL** (Area [10³ km²]: 2020, Range 68%, +1 m RSLR, Range 68%, Increase 2020-2100 [%]; Population [million]: 2020, Range 68%, +1 m RSLR, Range 68%, Increase 2020-2100 [%], 2000-20 pop. growth [$yr^{-1}$]).

| | 2m Area 2020 | % country area | 2m Area Range 68% | 2m Area +1 m RSLR | 2m Area +1m Range 68% | 2m Area Inc. 2020-2100 [%] | 2m Pop 2020 | 2m Pop Range 68% | 2m Pop +1 m RSLR | 2m Pop +1m Range 68% | 2m Pop Inc. 2020-2100 [%] | 2m Pop growth [$yr^{-1}$] | 0m Area 2020 | 0m Area Range 68% | 0m Area +1 m RSLR | 0m Area +1m Range 68% | 0m Area Inc. 2020-2100 [%] | 0m Pop 2020 | 0m Pop Range 68% | 0m Pop +1 m RSLR | 0m Pop +1m Range 68% | 0m Pop Inc. 2020-2100 [%] | 0m Pop growth [$yr^{-1}$] |
|---|---|---|---|---|---|---|---|---|---|---|---|---|---|---|---|---|---|---|---|---|---|---|---|
| Indonesia | 118.2 | 6.3 | 82-147 | 171.7 | 147-196 | 45 | 17.2 | 14-23 | 28.3 | 23-31 | 65 | 1.4 | 11.9 | 5-25 | 48.2 | 25-82 | 304 | 1.7 | 0-5 | 9.6 | 5-14 | 452 | 1.2 |
| United States | 100.3 | 1.1 | 83-116 | 129.6 | 116-144 | 29 | 4.2 | 3-7 | 8.2 | 7-10 | 97 | 0.5 | 14.3 | 4-42 | 63.7 | 42-83 | 345 | 0.6 | 0-1 | 2.2 | 1-3 | 285 | -0.9 |
| Brazil | 79.3 | 0.9 | 53-100 | 118.0 | 100-132 | 49 | 3.2 | 2-4 | 5.1 | 4-6 | 60 | 1.6 | 4.3 | 2-12 | 27.0 | 12-53 | 529 | 0.4 | 0-1 | 1.3 | 1-2 | 194 | 1.7 |
| Vietnam | 52.6 | 16.1 | 47-56 | 58.0 | 56-60 | 10 | 32.3 | 27-36 | 37.7 | 36-40 | 17 | 0.6 | 6.2 | 3-16 | 32.6 | 16-47 | 422 | 4.5 | 2-9 | 18.1 | 9-27 | 304 | 0.0 |
| Russian Federation | 51.3 | 0.3 | 34-71 | 91.4 | 71-110 | 78 | 0.2 | 0-0 | 1.2 | 0-1 | 449 | -0.5 | 3.3 | 1-8 | 19.0 | 8-34 | 468 | 0.0 | 0-0 | 0.1 | 0-0 | 1152 | -0.1 |
| China | 43.0 | 0.5 | 25-64 | 81.9 | 64-97 | 91 | 39.9 | 21-67 | 95.9 | 67-118 | 141 | 1.5 | 2.7 | 1-6 | 13.5 | 6-25 | 407 | 2.4 | 1-6 | 12.0 | 6-21 | 408 | 1.9 |
| Mexico | 39.5 | 2.0 | 34-45 | 50.3 | 45-56 | 27 | 0.7 | 1-1 | 1.1 | 1-1 | 53 | 1.2 | 3.9 | 1-15 | 25.3 | 15-34 | 545 | 0.0 | 0-0 | 0.3 | 0-1 | 854 | 0.7 |
| India | 38.6 | 1.2 | 25-47 | 55.3 | 47-61 | 43 | 38.7 | 25-45 | 51.2 | 45-57 | 32 | 0.9 | 3.8 | 2-7 | 14.2 | 4-22 | 278 | 4.0 | 2-7 | 13.4 | 7-25 | 234 | 0.3 |
| Venezuela | 30.3 | 3.3 | 22-36 | 39.0 | 36-41 | 29 | 0.7 | 1-1 | 1.0 | 1-1 | 33 | 4.6 | 1.7 | 0-4 | 11.6 | 6-21 | 572 | 0.1 | 0-0 | 0.1 | 0-1 | 224 | 3.3 |
| Australia | 30.0 | 0.4 | 21-39 | 48.0 | 39-56 | 60 | 0.3 | 0-1 | 0.8 | 1-1 | 191 | 1.7 | 3.6 | 3-6 | 11.5 | 4-13 | 223 | 0.0 | 0-0 | 0.1 | 0-0 | 355 | 1.7 |
| Myanmar | 22.8 | 3.4 | 13-31 | 37.2 | 31-42 | 63 | 3.9 | 2-6 | 9.4 | 6-13 | 143 | 0.7 | 3.6 | 3-4 | 7.2 | 2-15 | 101 | 0.0 | 0-0 | 0.9 | 0-2 | 312 | -0.6 |
| Nigeria | 22.7 | 2.5 | 21-24 | 25.7 | 24-27 | 13 | 11.9 | 11-13 | 13.4 | 13-15 | 13 | 3.4 | 1.5 | 1-13 | 19.2 | 13-21 | 11 | 1.4 | 1-6 | 9.8 | 6-11 | 596 | 3.3 |
| Bangladesh | 22.0 | 16.0 | 15-26 | 29.3 | 26-33 | 33 | 18.1 | 19-23 | 28.0 | 23-37 | 54 | 0.2 | 0.4 | 0-2 | 6.2 | 2-15 | 157 | 0.4 | 0-2 | 4.9 | 2-12 | 1151 | 0.3 |
| Thailand | 20.1 | 3.9 | 16-24 | 26.8 | 24-28 | 33 | 21.5 | 19-23 | 23.3 | 23-24 | 9 | 3.4 | 1.2 | 0-4 | 9.8 | 4-16 | 688 | 1.1 | 1-2 | 10.3 | 2-19 | 810 | 2.6 |
| Canada | 18.4 | 0.2 | 15-21 | 24.2 | 21-28 | 32 | 0.3 | 0-0 | 0.4 | 0-0 | 30 | 1.7 | 2.9 | 1-6 | 10.8 | 6-15 | 275 | 0.0 | 0-0 | 0.1 | 0-0 | 1258 | 1.8 |
| Denmark + Greenl. | 18.0 | 6.1 | 17-19 | 19.3 | 19-20 | 8 | 0.3 | 0-1 | 0.5 | 1-1 | 79 | 1.3 | 8.5 | 5-12 | 15.0 | 12-17 | 76 | 0.0 | 0-1 | 0.1 | 0-0 | 283 | -0.5 |
| Netherlands | 17.0 | 48.5 | 17-18 | 18.1 | 18-19 | 7 | 9.0 | 9-9 | 9.6 | 9-10 | 7 | 0.5 | 10.9 | 8-13 | 15.2 | 13-17 | 40 | 6.1 | 5-7 | 8.3 | 7-9 | 36 | 0.5 |
| Malaysia | 16.9 | 5.1 | 12-22 | 26.8 | 22-31 | 58 | 4.2 | 3-5 | 6.3 | 5-7 | 50 | 1.6 | 3.0 | 1-5 | 7.9 | 5-12 | 168 | 0.3 | 0-1 | 1.4 | 1-3 | 384 | 2.5 |
| Papua New Guinea | 13.2 | 2.9 | 9-17 | 20.6 | 17-24 | 55 | 0.1 | 0-0 | 0.2 | 0-0 | 74 | 1.7 | 1.4 | 1-3 | 5.7 | 3-9 | 314 | 0.0 | 0-0 | 0.1 | 0-0 | 332 | 1.6 |
| Suriname | 12.2 | 8.4 | 9-15 | 17.1 | 15-18 | 40 | 0.3 | 0-0 | 0.4 | 0-0 | 13 | 0.6 | 0.4 | 0-3 | 5.8 | 3-9 | 1361 | 0.0 | 0-0 | 0.2 | 0-0 | 738 | 8.4 |
| Tropics | 649 | 1.3 | 478-784 | 895 | 784-993 | 38 | 191 | 146-228 | 261 | 228-292 | 36 | 1.4 | 61 | 26-156 | 300 | 156-478 | 392 | 18 | 9-46 | 92 | 46-146 | 404 | 1.0 |
| [% World] | 62.1 | | 61-62 | 61.1 | 62-61 | | 71.7 | 74-67 | 63.6 | 67-62 | | | 46.5 | 43-53 | 58.0 | 53-61 | | 52.7 | 48-65 | 71.2 | 65-74 | | |
| Tropical America | 210 | 1.4 | 156-253 | 287 | 253-314 | 36 | 8 | 6-10 | 12 | 10-15 | 48 | 1.4 | 16.0 | 5-48 | 95.0 | 48-156 | 498 | 1.0 | 1-2 | 4.0 | 2-6 | 317 | 1.7 |
| [% World] | 20.1 | | 20-20 | 19.6 | 20-19 | | 3.1 | 3-3 | 3.0 | 3-3 | | | 12.2 | 8-16 | 18.4 | 16-20 | | 2.7 | 1-2 | 3.0 | 3-3 | | |
| Tropical Africa | 97 | 0.4 | 76-113 | 125 | 113-135 | 29 | 26 | 23-29 | 31 | 29-34 | 18 | 3.8 | 10.0 | 6-34 | 58.0 | 34-76 | 487 | 2.0 | 1-11 | 19.0 | 11-23 | 670 | 4.0 |
| [% World] | 9.3 | | 10-9 | 8.5 | 9-8 | | 9.9 | 11-8 | 7.6 | 8-7 | | | 7.5 | 9-12 | 11.2 | 12-10 | | 7.0 | 6-15 | 14.4 | 15-11 | | |
| Tropical Asia | 321 | 4.7 | 232-388 | 447 | 388-500 | 39 | 157 | 117-189 | 217 | 189-243 | 39 | 1.1 | 35.0 | 16-72 | 141.0 | 72-232 | 306 | 15.0 | 7-33 | 70.0 | 33-117 | 366 | 0.6 |
| [% World] | 30.6 | | 30-31 | 30.5 | 31-30 | | 58.7 | 60-55 | 53.0 | 55-51 | | | 26.5 | 26-25 | 27.2 | 25-30 | | 43.0 | 39-47 | 53.8 | 47-60 | | |
| Tropical Oceania | 22 | 0.7 | 14-29 | 37 | 29-44 | 70 | 0 | 0-0 | 0 | 0-0 | 91 | 0.8 | 0.0 | 0-2 | 6.0 | 2-14 | 1133 | 0.0 | 0-0 | 0.0 | 0-0 | 504 | 0.9 |
| [% World] | 2.1 | | 2-2 | 2.5 | 2-3 | | 0.0 | 0-0 | 0.0 | 0-0 | | | 0.4 | 0-1 | 1.1 | 1-2 | | 0.0 | 0-0 | 0.0 | 0-0 | | |
| Temperate & boreal | 397 | 0.4 | 306-487 | 569 | 487-647 | 43 | 75 | 51-112 | 149 | 112-181 | 98 | 1.1 | 70.0 | 35-137 | 217.0 | 137-306 | 210 | 16.0 | 10-25 | 37.0 | 25-51 | 127 | 1.0 |
| [% World] | 37.9 | | 39-38 | 38.9 | 38-39 | | 28.3 | 26-33 | 36.4 | 33-38 | | | 53.5 | 57-47 | 42.0 | 47-39 | | 47.3 | 52-35 | 28.8 | 35-26 | | |
| World | 1046 | 0.7 | 784-1271 | 1464 | 1271-1641 | 40 | 267 | 197-341 | 410 | 341-473 | 54 | 1.3 | 131 | 61-293 | 518 | 293-784 | 295 | 35 | 18-71 | 129 | 71-197 | 273 | 1.0 |

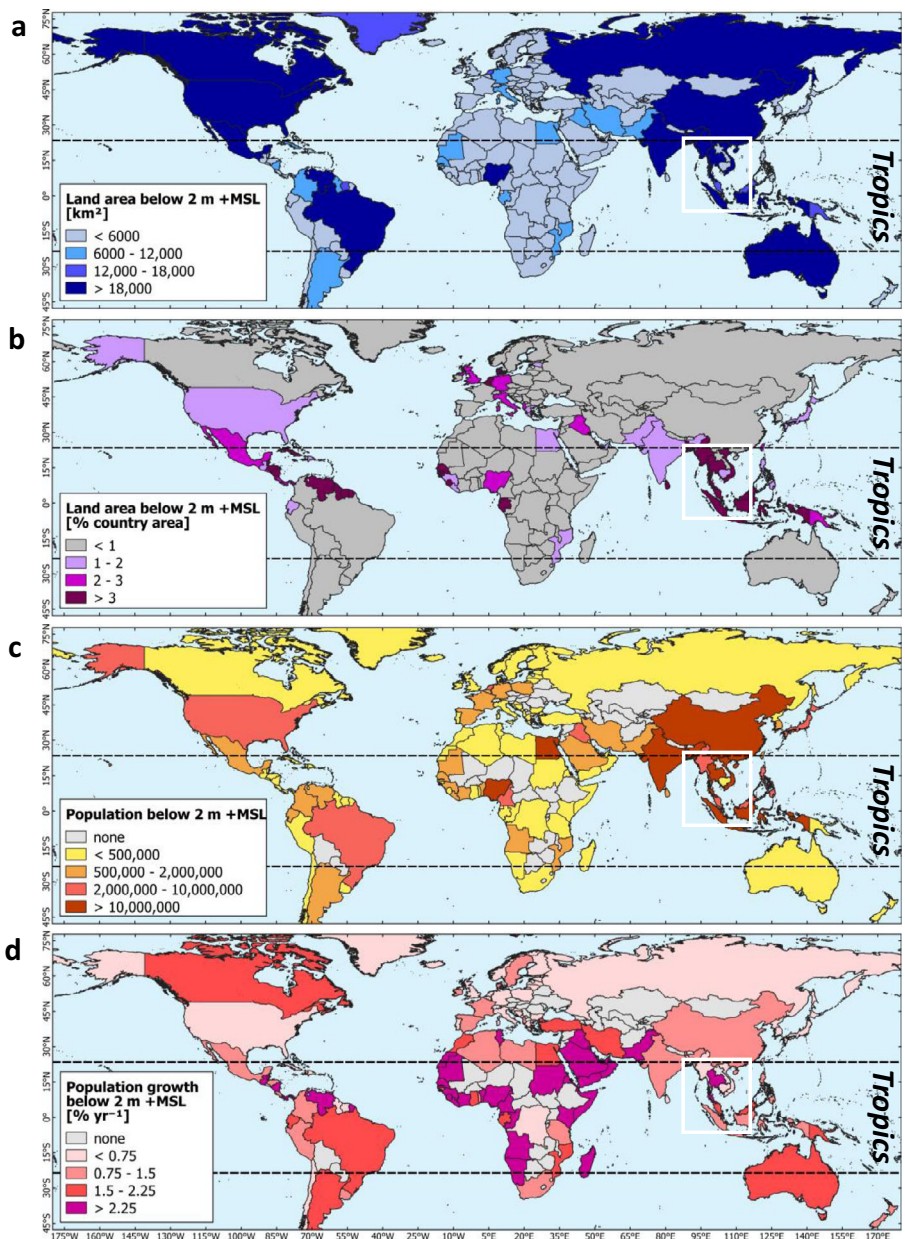

**Fig. 1 Global distribution of countries with lowland area below 2 m above mean sea level (+MSL) and population. a** Absolute and **b** relative coastal land areas, **c** population size and **d** population growth. Lowland elevation of the tropical Asia central region as indicated in white box is shown in Fig. 2.

For some tropical areas the resulting high flood risk is well understood, such as the Ganges-Brahmaputra-Meghna delta where about 20 to 60% of land is already flooded every year affecting tens of millions of people with hundreds of thousands of lives lost historically to cyclone related flooding[21]. However, the problem here is often seen as one of drainage congestion requiring improved infrastructure to allow water to flow from the land by gravity. This may be partly explained by the notion that much of the delta is still well above MSL, as suggested by most existing GDEMs. However, our new data reveal that in Bangladesh alone, land below 2 m +MSL covers 16% of its 2020 land area at 22,000 km² with a population of 18.1 million, while SRTM data yield only 1300 km² of land below 2 m +MSL and no other GDEM presents more than 13,900 km² (Supplementary Data). With only 1 m of RSLR, 6000 km² and 4.9 million people would be below MSL. This scenario appears highly likely as according to Becker et al.[21], RSLR in this area will reach 0.85–1.4 m by 2100

even under a greenhouse gas emission mitigation scenario (RCP4.5; ref. [1]), with LSS doubling the effect of SLR.

In another example of low awareness of the actual distribution of coastal flood risk, we find Indonesia to have the largest extent of land below 2 m +MSL of any country globally at 118,200 km² or 6.3% of its land area and 11.3% of the global total, 14 times more than the 8100 km² that is found using SRTM (Supplementary Data). Yet there is limited attention for sea-level rise vulnerability outside of a few urban areas, and the country is not usually prioritized in discussions of areas most at risk of SLR.

Given the rapidly increasing flood risk in extensive areas of tropical coastal lowlands below 0 and 2 m +MSL, there is no time to waste in developing adaptation measures. This will require spatial planning with a long-term perspective on flood risk based on accurate DTMs. The recent availability of satellite LiDAR data with global coverage can help to improve readiness to cope with SLR and LSS especially in those regions that to date were lacking accurate

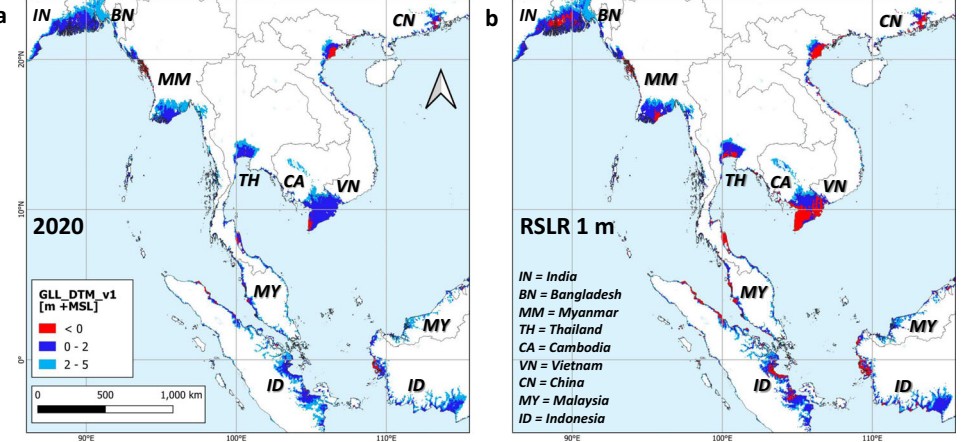

**Fig. 2 Lowland elevation in the central part of tropical Asia. a** In 2020 and **b** after 1 m relative sea-level rise (RSLR), as determined from GLL_DTM_v1. Details of Bangladesh, Thailand and Vietnam are shown in Fig. 4.

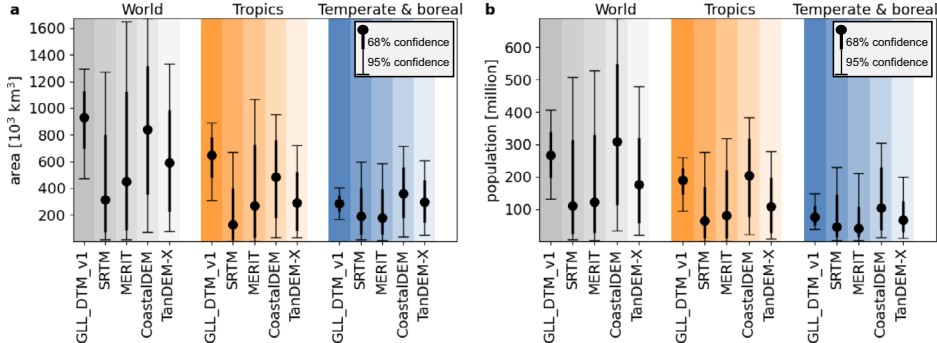

**Fig. 3 Coastal lowland area and population below 2 m above mean sea level (+MSL), with confidence levels at 68 and 95%; summary of Supplementary Data. a** Areas calculated from different Global Digital Elevation Models (SRTM90[13], MERIT[14], CoastalDEM[15] and TanDEM-X[23]) compared to GLL_DTM_v1[8] at 0.05-degree resolution, for countries ranked by land area greater than 12,000 km² below 2 m +MSL. For consistency, numbers are calculated within the SRTM coverage extent, between 60 N and 56 S. **b** Population data from ref. [22].

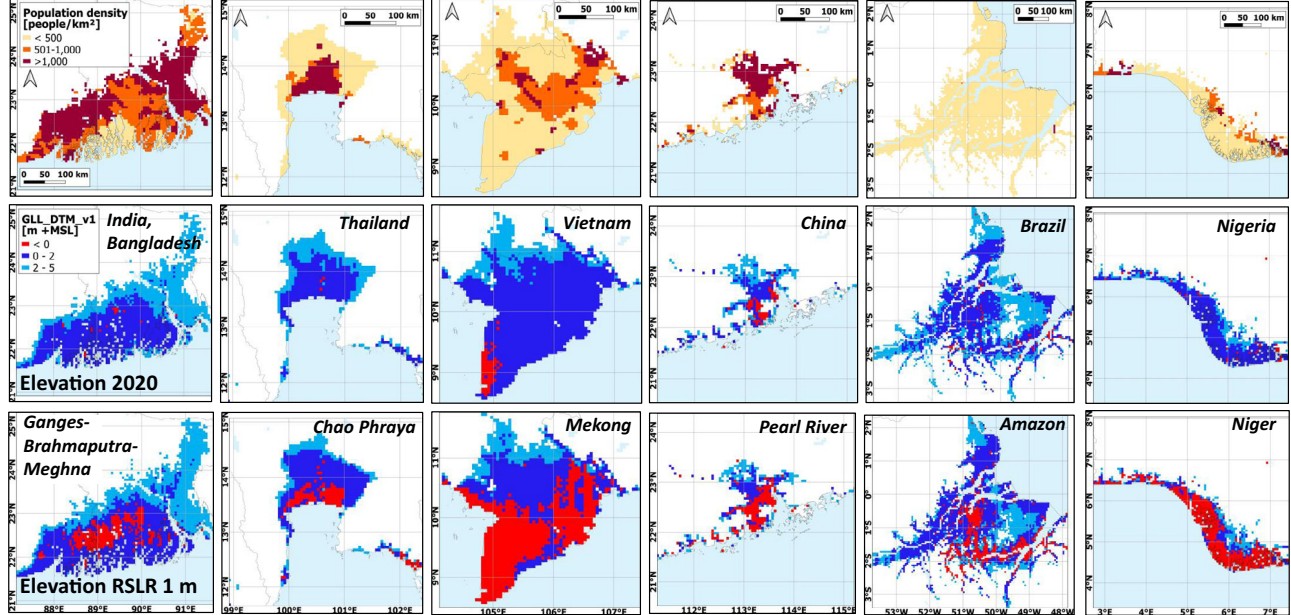

**Fig. 4 Population and land surface elevation below 5 m above mean sea level (+MSL) for 6 selected large tropical deltas.** (Top) Population in 2020 (middle) land surface elevation in 2020 and (bottom) land surface elevation after 1 m relative sea-level rise (RSLR).

DTMs to support adequate responses. To help make global LiDAR based DTMs more useful for spatial planning and policy making, further reduction in uncertainties and increase in resolution is ongoing as collection of satellite LiDAR data continues.

## Methods

**Elevation dataset**. The global LiDAR lowland DTM (GLL_DTM_v1) at 0.05-degree resolution (~5 × 5 km) is created from ICESat-2 data collected between 14 October 2018 and 13 May 2020[8].

**Coverage of analysis**. Global area and population numbers were calculated over the entire GLL_DTM_v1 extent of 88N-88S. Tropical numbers were calculated between 23.5N-23.5 S (Fig. 1).

**Definition of coastal lowland at highest risk of flooding**. We follow the definition by Syvitski et al.[17] who in a global review found coastal land below 2 m +MSL to be generally most susceptible to occasional river floods and storm surges, globally. In much of the World, such land is below common high tide sea levels and river flood levels.

**Current and recent coastal lowland population distribution**. Global population distribution in 2000 and 2020 was determined from the UN adjusted Gridded Population of the World database[22]. The population growth was determined from the trend between 2000 and 2020, for grid cells below 0 and 2 m +MSL.

**Relative sea-level rise (RSLR)**. We have estimated the land areas flooded and populations affected in future with a relative sea-level rise (RSLR) of 1 meter by 2100, which results in more or less equal parts from absolute SLR and land surface subsidence (LSS).

The range of SLR following from the IPCC RCP2.6/RCP8.5 projections by 2100 is 0.29–0.59/0.61–1.1 since 1986–2005 (ref. 1), from which we applied a middle value of 0.5 m rise since 2020.

The range of LSS rates as shown in Supplementary Table 1, of 2.5 to 10 mm yr$^{-1}$ in rural areas, also justify a middle value of 0.5 m between 2020 and 2100. Subsidence causes mentioned in sources are deforestation, drainage, groundwater abstraction and exploitation of oil and gas. Higher values are reported for urban areas, often well in excess of 20 mm yr$^{-1}$, but not applied in this assessment to maintain global uniformity in parameters.

**Uncertainty assessment**. Estimates of area and population currently below 0 and 2 m +MSL and following a RSLR of 1 m were calculated using two methods. The deterministic method, often used in flood risk assessments but providing no indication of quality, and the modified deterministic method which takes into account vertical uncertainty of the elevation model[16]. Given that GLL_DTM_v1 for land below 2 m +MSL has an RMSE of 0.5 m[8] and we apply a RSLR of 1 m, areas can be estimated at 68% confidence level, i.e. projected areas fall somewhere within the stated range. For 68% confidence, the lower end of the range is at 0.5 m elevation (1 m RLSR − 0.5 m RMSE) and the upper end of the range is at 1.5 m elevation (1 m RLSR + 0.5 m RMSE).

Similarly, areas and population currently below 0 and 2 m +MSL at both 68 and 95% (where range is ±1.96 x RMSE) confidence levels were calculated in comparison with other GDEMs for which RMSE's were assessed by Vernimmen et al.[8] at 0.05-degree resolution.

## Data availability

The GLL_DTM_v1 dataset applied in this study is available online at https://doi.org/10.17632/v5x4vpnzds.1. The global population dataset is available from https://doi.org/10.7927/H45Q4T5F.

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

## Author contributions

A.H. and R.V. were both involved in all stages and aspects of the work.

## Competing interests

The authors declare no competing interests.
