## [Peer Review File · Nature Communications]

REVIEWER COMMENTS

Reviewer #1 (Remarks to the Author):

This paper presents an analysis of the global risk of sea level rise impacts based on a new higher accuracy global lowland DEM derived from satellite lidar data. The main findings are that the potential impacts of sea-level rise are greatest in the tropics, and that use of the improved DEM has allowed better estimates of impacts that are substantially above estimates from lower accuracy DEMs.

High quality estimates of the global area and corresponding resources, especially population, subject to adverse effects of sea level rise are needed. The quality of the elevation data used in such analyses is crucial (this has been shown in numerous studies), thus the use of an improved DEM here is an advance, especially as comparisons are presented with the other global DEMs that have been widely used. However, the study suffers from the same deficiency of many previous similar studies in that there is no information about the quality of the reported results (area and population at risk), namely what is the confidence level of the reported numbers. Yes, the new global lowland DEM used here is superior to other DEMs in terms of absolute vertical accuracy (shown clearly in the previous 2020 publication on the DEM by the same authors), and the results presented in this paper show that they differ from results when using previous global DEMs, but the reader has to just trust that the results are good because the underlying DEM is presumably better. The argument of improved estimates would be bolstered substantially by quantifying and reporting the confidence of the results, and methods exist to do that.

First, the validity of using GLL_DTM_v1 for the analysis of 1 m of RSLR can be quantified with a method described in (Gesch, 2018). As shown in the authors previous 2020 paper in Remote Sensing, GLL_DTM_v1 has a 0.5 m RMSE for areas <2 m in elevation. Comparatively, the other global DEMs (SRTM90, MERIT, CoastalDEM, and TanDEM-X) all exhibit a vertical accuracy in the <2 m elevation zone that is well over 1 m RMSE, so the inherent vertical error in each case is more than the elevation increment (1 m RSLR) that is being mapped. So, there is a clear advantage to using GLL_DTM_v1, and it can be quantified. Using the approach in (Gesch, 2018), it can be shown that GLL_DTM_v1 supports mapping an elevation interval of 1 m at a confidence level of 68%. Because the other global DEMs have much worse vertical accuracy, the calculated confidence level for each for mapping a 1 m RLSR would be much worse.

Second, the quality of the reported results (area and population potentially impacted by a 1 m RLSR) could be improved substantially by associating a confidence level. The approach used in this study is the simple deterministic method often employed in sea level rise assessments, and consequently there is no indication of the quality of the reported results. The preferred fully probabilistic approach would have quantitative information on the confidence (or likelihood) of the results. An approach that does provide some indication of the quality (confidence) of the results that would be appropriate for a global scale study such as this one is the modified deterministic approach described in (Gesch, 2018). For this study, the modified deterministic approach would make use of the known vertical accuracy of GLL_DTM_v1 (0.5 m RMSE) to provide a range of values (for impacted area and population) in contrast to just a single value from the deterministic approach, and this range of values would have a specific confidence level associated with it. For 68% confidence, the lower end of the range would be at 0.5 m elevation (1 m RLSR – 0.5 m RMSE) and the upper end of the range would be at 1.5 m elevation (1 m RLSR + 0.5 m RMSE). For 95% confidence, the vertical error term used would be 0.5 m x 1.96, or 0.98 m. Doing the same modified deterministic approach for the other global DEMs (using the corresponding vertical accuracy for each) would still show GLL_DTM_v1 to be superior as the ranges of values would be much smaller at a given confidence level.

In addition to needed improvements in quantifying the quality of reported results, some other items in the manuscript need to be addressed. Please see the annotated manuscript for comments tied to specific locations in the paper.

Reviewer #2 (Remarks to the Author):

Review of Hooijer and Vernimmen "New land elevation data reveal greatest sea-level rise impact in the tropics"

The issue of inaccuracy and uncertainty in land elevation is a significant issue in understanding the risks of sea-level rise and marine hazards in general, which has been widely discussed in recent years. Depending on the sources used very different results can be produced. Hence, this paper being focussed on a new and more accurate global elevation data set is of great interest and this work has the potential to attract great interest. The paper reads well and makes sense, but on reflection there are a number of issues that need to be carefully considered before it is published as it also raises questions.

The overarching point about the tropics being affected is not surprising looking back at earlier results which have widely identified south, south-east and east Asia as the global hotspot for sea-level rise impacts and Africa following behind (e.g. Nicholls and Cazenave, 2010, *Science* – but there are many other papers). So is this a new result or rather a confirmation of existing results in more detail?

Reference to wider literature on DEMs and standards that should be followed is needed -- e.g., Gesch (2018) doi: 10.3389/feart.2018.00230

Table 1 is interesting but hard to follow – I think it shows the land area and population below mean sea level and below 2 m above mean sea level but I do not understand many of the columns. The Table needs simplification and/or the caption needs expansion. For example, the caption only mentioned below 2 m above mean sea level. Are these numbers credible as the people below mean sea level will depend on some form of defence to live there – probably dikes and polders. Are 1.4 million people really living below sea level/behind defences in Nigeria. I have never been to Nigeria but I am not aware of significant defence systems so I am a little sceptical of this result. Are there other countries where similar questions are raised?

Table 1 is very nice as I am not aware of any earlier good estimates of population below mean sea level – so this should be kept subject to addressing my queries.

Figure 2 and in fact all the results with SLR – it is odd that the study area is not expanded as in addition to the area below 2 m becoming relatively lower the area within 2 m of sea level expands. Why is this not considered – I can see it is more work but a clear statement about this would be useful.

Line 27-30 – I broadly agree with this but find the statement a little sweeping without any reference to support. What is the source of these numbers? And are all coastal lowlands subsiding – certainly sedimentary lowlands are subsiding, but for example the US East Coast is experiencing loss of elevation, but due to Glacial Isostatic Adjustment.

Line 91-92 – no population growth after 2020 is not really a conservative assumption – all analyses to 2100 expect population growth – I am not saying change but consider your motivation for this assumption – which I take as ease of calculation?

Line 112 – deaths in cyclone impact happened historically but strides in cyclones warnings and shelters have reduced the death toll dramatically. This was apparent in Cyclone Sidr in 2007 and Cyclone Amphan in 2020. Your text should acknowledge this.

Line 121-127 – and Indonesia is on the equator so not so subjected to tropical cyclones as is the Philippines to the north. I agree it is an interesting result, but how much does tidal and surge regime influence your results?

Line 130 – and widely in Asia – dikes and polders are widespread in China, Vietnam and Bangladesh

Line 132-134 – this suggests little understanding of the situation in Asia – there are 5,000 km of dike in coastal Bangladesh and the last remaining unpoldered area will be poldered shortly. And there are better extensive and better dike systems in China and Vietnam – and major dikes were constructed in Thailand after the 2011 floods.

Line 134-137 – entirely speculative statement. You might be right but you do not know and defining the areas that are threatened are where these analyses should stop. Further the high population and

growing wealth in Asia means that a Dutch-type approach to adaptation may emerge. See for example Lincke and Hinkel, 2018, Global Environmental Change for a global coast benefit analysis. Line 138-143 – this is good and I agree.

RESPONSES TO REFEREE COMMENTS

EDITOR COMMENTS (BY EMAIL 5 MARCH 2021):

Editor comment: As you will see from the reports copied below, the reviewers raise important concerns. We find that these concerns limit the strength of the study, and therefore we ask you to address them with additional work. Without substantial revisions, we will be unlikely to send the paper back to review.

Author response: We acknowledge the validity of the Reviewers' comments and appreciate their constructive suggestions. We think we have comprehensively addressed all. Revisions have been substantial and have improved the article.

In particular, Reviewer #1 felt that confidence intervals are needed to ensure the robustness of your data and the improvement over existing methods, and Reviewer #2 suggested that some of the conclusions should be toned down and better contextualized with citations to past literature.

We think we have been able to address both issues, as we show in the response to Reviewers below. Confidence intervals have been determined in response to Reviewer #1's comments, following the suggested method. And conclusions / discussion pointed out by Reviewer #2 (and sometimes Reviewer #1) as requiring more work or justification have been adjusted, with the paragraph in Discussion of possible solutions to increased flood risk being removed altogether.

REVIEWER #1 (REMARKS TO THE AUTHOR):

Reviewer comment: This paper presents an analysis of the global risk of sea level rise impacts based on a new higher accuracy global lowland DEM derived from satellite lidar data. The main findings are that the potential impacts of sea-level rise are greatest in the tropics, and that use of the improved DEM has allowed better estimates of impacts that are substantially above estimates from lower accuracy DEMs.

High quality estimates of the global area and corresponding resources, especially population, subject to adverse effects of sea level rise are needed. The quality of the elevation data used in such analyses is crucial (this has been shown in numerous studies), thus the use of an improved DEM here is an advance, especially as comparisons are presented with the other global DEMs that have been widely used. However, the study suffers from the same deficiency of many previous similar studies in that there is no information about the quality of the reported results (area and population at risk), namely what is the confidence level of the reported numbers. Yes, the new global lowland DEM used here is superior to other DEMs in terms of absolute vertical accuracy (shown clearly in the previous 2020 publication on the DEM by the same authors), and the results presented in this paper show that they differ from results when using previous global DEMs, but the reader has to just trust that the results are good because the underlying DEM is presumably better. The argument of improved estimates would be bolstered substantially by quantifying and reporting the confidence of the results, and methods exist to do that.

Author response: We agree that adding such statistics will strengthen the paper and welcome the Reviewer suggestions.

First, the validity of using GLL_DTM_v1 for the analysis of 1 m of RSLR can be quantified with a method described in (Gesch, 2018). As shown in the authors previous 2020 paper in Remote Sensing, GLL_DTM_v1 has a 0.5 m RMSE for areas <2 m in elevation. Comparatively, the other global DEMs (SRTM90, MERIT, CoastalDEM, and TanDEM-X) all exhibit a vertical accuracy in the <2 m elevation zone that is well over 1 m RMSE, so the inherent vertical error in each case is more than the elevation increment (1 m RSLR) that is being mapped. So, there is a clear advantage to using GLL_DTM_v1, and it can be quantified. Using the approach in (Gesch, 2018), it can be shown that GLL_DTM_v1 supports mapping an elevation interval of 1 m at a confidence level of 68%. Because the other global DEMs have much worse vertical accuracy, the calculated confidence level for each for mapping a 1 m RLSR would be much worse.

We thank the reviewer for this helpful suggestion. We have applied the suggested method, described it in the Method Section and added results for GLL_DTM_v1 at the confidence level of 68% to the main text and in Table 1, for both Area and associated Population. Results of comparison with GDEMs are presented in Supplementary Table 2 and new Supplementary Figure 1 to keep the main text focused and compact, and comply with the Nature requirements for text length and number of Tables/Figures.

Second, the quality of the reported results (area and population potentially impacted by a 1 m RLSR) could be improved substantially by associating a confidence level. The approach used in this study is the simple deterministic method often employed in sea level rise assessments, and consequently there is no indication of the quality of the reported results. The preferred fully probabilistic approach would have quantitative information on the confidence (or likelihood) of the results. An approach that does provide some indication of the quality (confidence) of the results that would be appropriate for a global scale study such as this one is the modified deterministic approach described in (Gesch, 2018). For this study, the modified deterministic approach would make use of the known vertical accuracy of GLL_DTM_v1 (0.5 m RMSE) to provide a range of values (for impacted area and population) in contrast to just a single value from the deterministic approach, and this range of values would have a specific confidence level associated with it. For 68% confidence, the lower end of the range would be at 0.5 m elevation (1 m RLSR – 0.5 m RMSE) and the upper end of the range would be at 1.5 m elevation (1 m RLSR + 0.5 m RMSE). For 95% confidence, the vertical error term used would be 0.5 m x 1.96, or 0.98 m. Doing the same modified deterministic approach for the other global DEMs (using the corresponding vertical accuracy for each) would still show GLL_DTM_v1 to be superior as the ranges of values would be much smaller at a given confidence level.

We agree and have calculated and added this information as requested, in Supplementary Table 2 comparing results for GLL_DTM_v1 and GDEMs. As the Table is very large and complex, we have also added a summary Supplementary Figure 1 that clearly shows the far narrower confidence range of results calculated with GLL_DTM_v1 as compared with the GDEMs.

In addition to needed improvements in quantifying the quality of reported results, some other items

in the manuscript need to be addressed. Please see the annotated manuscript for comments tied to specific locations in the paper.

We have taken the comments from the PDF text and placed below.

REVIEWER #1 – FURTHER COMMENTS TAKEN FROM PDF FILE:

Line 1 (Title) – This seems like a rather strong statement that could arguably be open to debate. Isn't it actually "potential" SLR impacts, or areas that are at risk (or susceptible to) SLR impacts. As written, it implies the impacts are a definite, whereas even the authors themselves (in lines 128-131) point out that the extent impacts can be greatly affected by coastal protection measures.

We agree and have replaced “New land elevation data reveal greatest sea-level rise impact in the tropics” by “New land elevation data reveal greatest sea-level rise vulnerability in the tropics”. As an alternative we propose “New land elevation data reveal greatest area and population vulnerable to sea-level rise is in the tropics” to make the type of vulnerability more explicit (area and population, not necessarily economic or ecosystem values), but this may somewhat exceed to number of characters allowed by Nature Communications. We leave it to the Editor whether this alternative can be used.

Line 41 – Would be good to also give RMSE, as that is most widely used DEM error metric.

We agree and have added the RMSE value. Leaving the mean absolute error in place as we find this metric is better understood by non-experts.

Line 46 – Would be good to also give RMSE, as that is most widely used DEM error metric.

We have added the RMSE value.

Line 67 – What is the source of population numbers? It should be cited.

We have added the source reference both here in Table 1 and in Supplementary Table 2 as well, which was before only noted in Methods.

Line 125-126 – This sentence seems to be conjecture. While it may be true that SRTM and other GDEMs indicate much smaller areas at risk, is that really the reason Indonesia gets only limited attention for flood problems and is not prioritized in discussions of coastal risk? Likely not. There shouldn't be a cause and effect relationship here. The only case where the statement in the manuscript might be true is if it could be shown (with references) that SRTM (or other GDEMs) were used in a study of Indonesia and risk was under represented. It is highly unlikely such a study exists.

We agree that this implied causality can not be proven based on scientific publications and we have adjusted the text to avoid the suggestion that it can be. However, as a background clarification to the Reviewer: from our ~20 years of advisory and research

work in Asia, assessing coastal lowland use optimization and defense options for Government and donor Banks, we have indeed found that flood risk continues to be systematically underestimated by Indonesian authorities. We have seen the higher elevations of the SRTM DEM being used in projects that then failed because of flooding. And SRTM data are still used in attempts to push through coastal development and protection projects that would be found to be unviable if better elevation data would be applied. This is one reason we have worked on improved DTMs, that until satellite data became available were mostly project based field and airborne LiDAR elevation surveys (see e.g. our paper <https://www.mdpi.com/2072-4292/11/10/1152>). Results of such advisory studies are not released to the public so can unfortunately not be quoted.

Line 134-135 – This statement should be supported with citation of a reference.

The Section where the statement was made was removed in its entirety, also in response to comments by Reviewer #2.

Line 142-143 – Although the coarse resolution of the DTM used in this study (about 5 km) would likely prove limiting for use in planning local adaptation measures. A comment about this should be added here.

We have added a comment as suggested.

Line 184 – Even though this is likely a well known document, more information on how to actually locate it is needed.

We have added a more complete reference.

REVIEWER #2 (REMARKS TO THE AUTHOR):

Review of Hooijer and Vernimmen “New land elevation data reveal greatest sea-level rise impact in the tropics”. The issue of inaccuracy and uncertainty in land elevation is a significant issue in understanding the risks of sea-level rise and marine hazards in general, which has been widely discussed in recent years. Depending on the sources used very different results can be produced. Hence, this paper being focussed on a new and more accurate global elevation data set is of great interest and this work has the potential to attract great interest. The paper reads well and makes sense, but on reflection there are a number of issues that need to be carefully considered before it is published as it also raises questions.

We thank the reviewer for the positive comments on interest and readability of the paper. We will address the issues raised in our responses below.

The overarching point about the tropics being affected is not surprising looking back at earlier results which have widely identified south, south-east and east Asia as the global hotspot for sea-level rise impacts and Africa following behind (e.g. Nicholls and Cazenave, 2010, Science – but there are many other papers). So is this a new result or rather a confirmation of existing results in more detail?

We acknowledge that it is not entirely new information that large tropical coastal land areas in especially SE Asia have high flood risk. We are not aware however of a published global assessment that quantifies both absolute and relative areas and populations at most immediate risk as we have done, and on that basis finds the majority of such land to be in the tropics or specifically in tropical Asia. To our knowledge no other global study has looked at land area and population below sea level. Indeed, our assessment of land and population below 0 and 2 m +MSL would have been impossible to date, in the absence of global coastal elevation data with sufficient accuracy at the lowest elevations, which as Reviewer #1 (referring to Gesch, 2018) correctly observes is not possible with an RMSE over 1 m as existing GDEMs have. Earlier publications were necessarily somewhat more general, sometimes somewhat qualitative. Following suggestions by Reviewer #1 we have added analysis of confidence levels, in main text and Supporting Information, that better brings out the novelty of our results.

Apart from technicalities, we can say it has certainly surprised us and our direct colleagues working in this field to find numbers as high as 62% of the area and 72% of population at highest risk being in the tropics, and 59% of population in tropical Asia alone. The presence of so much very low-lying land in tropical America (double the area in tropical Africa), that could not be mapped to date as it is still largely forested, has also been a surprise to us and our colleagues working in that region. In this context, we have added a map of Amazon Delta elevation to Figure 3, and removed the map of the Nile Delta (which is actually outside of the tropical zone, even though southern Egypt is in it).

We have added references to earlier studies (Syvitski et al., 2009; Nicholls and Cazenave 2010) that indicate regions to be global flood vulnerability hotspots, although they do not provide or compare regional numbers of areas or populations at risk. We now also refer to the very recent Nicholls et al. (2021) paper that does have a more quantitative approach

but still uses the SRTM elevation data that is not accurate enough to identify areas most at risk, and therefore understandably presents data for areas below 10 m.

Reference to wider literature on DEMs and standards that should be followed is needed -- e.g, Gesch (2018) doi: 10.3389/feart.2018.00230

Reference to the standards proposed by Gesch (2018) was added, also in response to more detailed comments on confidence statistics by Reviewer #1 in this respect.

Table 1 is interesting but hard to follow – I think it shows the land area and population below mean sea level and below 2 m above mean sea level but I do not understand many of the columns. The Table needs simplification and/or the caption needs expansion. For example, the caption only mentioned below 2 m above mean sea level.

We have completely revised the Table for improved clarity, and adjusted the caption. We have also added 68% confidence to the table as well as breakdown for the tropical America / tropical Africa and tropical Oceania regions. In order to keep the table somewhat compact we have removed 2000 population.

Are these numbers credible as the people below mean sea level will depend on some form of defence to live there – probably dikes and polders. Are 1.4 million people really living below sea level/behind defences in Nigeria. I have never been to Nigeria but I am not aware of significant defence systems so I am a little sceptical of this result. Are there other countries where similar questions are raised?

This is a good point. Actually, we do find that ground levels in parts of tidal mangrove areas such as the Niger Delta (Nigeria) but also the Mekong Delta (Vietnam) and Indonesian coastlines are around and sometimes just below (decimetres) mean sea level according to GLL_DTM_v1. From our field observations in Asia (GBM Delta, Mekong Delta, Indonesian coastlines) this appears related to the presence of numerous fish ponds, often associated with local tidal water control structures (e.g. tidal flap gates). Although population densities in such mangrove / fish pond areas tend to be low, it can add up over large expanses. It should be noted of course that the people in such areas tend to live on small patches of somewhat higher ground, often too small to be picked up by DTMs.

Table 1 is very nice as I am not aware of any earlier good estimates of population below mean sea level – so this should be kept subject to addressing my queries.

Thank you. We have aimed to address the reviewer points in improving the table, and have expanded text on area and population below MSL throughout the paper.

Figure 2 and in fact all the results with SLR – it is odd that the study area is not expanded as in addition to the area below 2 m becoming relatively lower the area within 2 m of sea level expands.

Why is this not considered – I can see it is more work but a clear statement about this would be useful.

There appears to be a misunderstanding. The area <2m +MSL is in fact expanded as we apply 1 m RSLR, in Figure 2 and in all aspects of the analysis. However, this is better visible in Figure 3 because of the somewhat zoomed-in scale.

Line 27-30 – I broadly agree with this but find the statement a little sweeping without any reference to support. What is the source of these numbers? And are all coastal lowlands subsiding – certainly sedimentary lowlands are subsiding, but for example the US East Coast is experiencing loss of elevation, but due to Glacial Isostatic Adjustment.

Table 1 in Supplementary Information provides a sample of subsidence numbers for literature including for US East Coast. We are aware that more references can be found but we have not aimed for an exhaustive review here, if only because of the Nature Communication limitation on the number of references allowed.

Line 91-92 – no population growth after 2020 is not really a conservative assumption – all analyses to 2100 expect population growth – I am not saying change but consider your motivation for this assumption – which I take as ease of calculation?

The main reason for making this assumption is that the alternative assumption, of a growing coastal lowland population, would require an undisputed source whereas we find there is much uncertainty around this topic certainly when it comes to coastal urban populations that tend to be highly affected by migration and economic factors. In fact, we originally did a scenario analysis following the five coastal population SSPs proposed by Merkens et al. 2016, which we could provide to the Reviewer separately if requested, but were confounded by the outcome as these scenarios actually predict coastal population shrinkage to 2020 numbers by 2100 after a peak around 2060. We agree with the reviewer that coastal population growth by 2100 seems more likely but as this is not clear from suitable sources we decided to not include the work on population projections. We have added a sentence on population growth to Discussion, including a reference to Merkens et al. 2016.

Line 112 – deaths in cyclone impact happened historically but strides in cyclones warnings and shelters have reduced the death toll dramatically. This was apparent in Cyclone Sidr in 2007 and Cyclone Amphan in 2020. Your text should acknowledge this.

We have added the word ‘historically’.

Line 121-127 – and Indonesia is on the equator so not so subjected to tropical cyclones as is the Philippines to the north. I agree it is an interesting result, but how much does tidal and surge regime influence your results?

We acknowledge that the processes involved in coastal extreme high-water events vary across regions, and often multiple processes are involved. While much of Indonesia

(except Aceh and North Sulawesi) is currently outside the tropical cyclone belt, it is highly prone to tsunamis (the 2004 event took over 220,000 lives in Indonesia alone) and in the Java Sea to extreme high tides caused by complex hydrodynamics that have caused catastrophic flooding in major cities like Jakarta and Banjarmasin. Moreover, climate predictions for the country include increased cyclone activity that may have already started in Java (several unusually severe storms occurred in recent years, from the southern cyclone system). In this paper we do not aim to discuss flooding processes, for the sake of compactness.

Line 130 – and widely in Asia – dikes and polders are widespread in China, Vietnam and Bangladesh

We have carefully considered the Reviewer comments on the current flood protection situation in tropical Asia and future developments, be it towards a traditional ‘Dutch polder’ system or towards the more resilient ‘building with nature approach’ that is now developed in the Netherlands. A more resilient approach is currently also considered in GBM and Mekong deltas where some dikes were actually removed because they are not always considered a sustainable solution in the face of sea-level rise and subsidence that is partly attributed to sediment starvation caused by dikes. This is a complex discussion that should either be explored thoroughly or, as the Reviewer suggests, left alone. We decided to do the latter, and have removed all text on the topic of response and adaptation from the paper, also to keep the text within the length allowed by Nature.

Line 132-134 – this suggests little understanding of the situation in Asia – there are 5,000 km of dike in coastal Bangladesh and the last remaining unpoldered area will be poldered shortly. And there are better extensive and better dike systems in China and Vietnam – and major dikes were constructed in Thailand after the 2011 floods.

Understanding the situation may not be the issue here as both authors have lived and worked in Asia for most of the last 20 years, often around coastal land and water management issues including feasibility assessments for coastal protection schemes in mostly Indonesia and Vietnam (these are unpublished as conducted for Governments and Banks). However, there may be some differences in interpretation and focus, as this is a complex topic. Therefore, we have removed this text to maintain a focus in this paper on the technical analysis.

Line 134-137 – entirely speculative statement. You might be right but you do not know and defining the areas that are threatened are where these analyses should stop. Further the high population and growing wealth in Asia means that a Dutch-type approach to adaptation may emerge. See for example Lincke and Hinkel, 2018, Global Environmental Change for a global coast benefit analysis.

We have removed this text as we agree with the Reviewer that it suffices to have an analysis of threatened areas without discussing responses and consequences.

Line 138-143 – this is good and I agree.

Thank you.

REVIEWERS' COMMENTS

Reviewer #1 (Remarks to the Author):

Thank you for the detailed responses to the review comments. All the comments have been suitably addressed, and I believe the manuscript and findings have been improved substantially. I recommend the manuscript be accepted for publication.

Reviewer #2 (Remarks to the Author):

I am pleased with the author's response to my earlier. I think the paper is ready to be published. There are a few comments to think about.

One big comment that the authors can consider is current rates of Land Surface Subsidence means that 1 m relative sea-level rise will be exceeded much earlier than 2100 in some locations -- adds to the immediacy of the results.

Why capitalise "Delta" -- I seek the advise of the editors but I would say "delta".

In Supplementary Table 1, Coastal Louisiana and the Mississippi delta are listed -- are they different as I see them as the same. Please review and tidy.

AUTHORS' RESPONSE TO FINAL REVIEWERS' COMMENTS

Reviewer #1 (Remarks to the Author):

Thank you for the detailed responses to the review comments. All the comments have been suitably addressed, and I believe the manuscript and findings have been improved substantially. I recommend the manuscript be accepted for publication.

RESPONSE: thank you.

Reviewer #2 (Remarks to the Author):

I am pleased with the author's response to my earlier. I think the paper is ready to be published. There are a few comments to think about.

One big comment that the authors can consider is current rates of Land Surface Subsidence means that 1 m relative sea-level rise will be exceeded much earlier than 2100 in some locations -- adds to the immediacy of the results.

RESPONSE: our aim is to be conservative. We agree with the comment but do not think it warrants major text changes. The Introduction already indicated that 1 m RSLR may be reached before 2100, we have now reiterated this point in Discussion.

Why capitalise "Delta" -- I seek the advise of the editors but I would say "delta".

RESPONSE: we have changed 'Delta' to 'delta' throughout the text.

In Supplementary Table 1, Coastal Louisiana and the Mississippi delta are listed -- are they different as I see them as the same. Please review and tidy.

RESPONSE: actually the source publications describe somewhat different areas, with Coastal Louisiana extending beyond the Mississippi i.e. presenting a larger area. So we suggest the table is better left as is.